# Water Quality Assessment Bias Associated with Long-Screened Wells Screened across Aquifers with High Nitrate and Arsenic Concentrations

**DOI:** 10.3390/ijerph19169907

**Published:** 2022-08-11

**Authors:** Yibin Huang, Yanmei Li, Peter S. K. Knappett, Daniel Montiel, Jianjun Wang, Manuel Aviles, Horacio Hernandez, Itza Mendoza-Sanchez, Isidro Loza-Aguirre

**Affiliations:** 1Department of Geology & Geophysics, Texas A&M University, College Station, TX 77843, USA; 2Department of Mining, Metallurgy and Geology Engineering, University of Guanajuato, Guanajuato 36000, Mexico; 3Department of Geological Sciences, University of Alabama, Tuscaloosa, AL 35487, USA or; 4Geosyntec Consultants, Clearwater, FL 33764, USA; 5Three Gorges Geotechnical Engineering Co., Ltd., Wuhan 430074, China; 6Department of Geomatic and Hydraulic Engineering, University of Guanajuato, Guanajuato 36000, Mexico; 7Environmental and Occupational Health Department, Texas A&M University, College Station, TX 77843, USA

**Keywords:** long-screened wells, nitrate, arsenic, geogenic contaminants, geothermal, Mexico, semi-arid, arid, urban contaminants, agriculture contaminants

## Abstract

Semi-arid regions with little surface water commonly experience rapid water table decline rates. To hedge against the falling water table, production wells in central Mexico are commonly installed to depths of several hundred meters below the present water table and constructed as open boreholes or perforated casings across their entire length. Such wells represent highly conductive pathways leading to non-negligible flow across chemically distinct layers of an aquifer—a phenomenon known as ambient flow. The objectives of this study were to estimate the rate of ambient flow in seven production wells utilizing an end-member mixing model that is constrained by the observed transient chemical composition of produced water. The end-member chemical composition of the upper and lower layers of an urban aquifer that overlies geothermal heat is estimated to anticipate the future quality of this sole source of water for a rapidly growing urban area. The comprehensive water chemistry produced by seven continuously perforated municipal production wells, spanning three geologically unique zones across the city of San Miguel de Allende in Guanajuato State, was monitored during one day of pumping. The concentration of conservative constituents gradually converged on steady-state values. The model indicates that, relative to the lower aquifer, the upper aquifer generally has higher specific conductance (SC), chloride (Cl), nitrate (NO_3_), calcium (Ca), barium (Ba) and magnesium (Mg). The lower aquifer generally has a higher temperature, sodium (Na), boron (B), arsenic (As) and radon (Rn). Ambient flow ranged from 33.1 L/min to 225.7 L/min across the seven wells, but this rate for a given well varied depending on which tracer was used. This new 3D understanding of the chemical stratification of the aquifer suggests that as water tables continue to fall, concentrations of geothermally associated contaminants of concern will increase in the near future, potentially jeopardizing the safety of municipal drinking water.

## 1. Introduction

By 2025, approximately 70% of the world’s population will live in cities [1]. This will continue to drive urban water demand higher. Compared to rivers, lakes and reservoirs, aquifers store vastly more water and are less vulnerable to surface contamination. For these reasons, groundwater is the main source of potable water for many of the world’s megacities such as Dhaka, Mexico City and Hong Kong [2]. Although relatively protected from surface contamination, spatially concentrated municipal pumping can still lead to deterioration of water quantity and quality [3,4,5]. Water table declines can be dramatic in semi-arid and arid regions. For example, in Irbid, Jordan, municipal and irrigation pumping from groundwater increased 227% over the last 40 years driving the water table 70 m lower [6]. Similarly, in the city of Torreon, Mexico, pumping for agriculture and municipal supply caused water levels to decline by 170 m over the past 100 years [7]. In the coastal city of Jakarta, the capital of Indonesia, intensive municipal pumping caused groundwater levels to decline by 20–25 m in the city center and deeper in areas to the west and east [8]. This drove substantial land subsidence at rates ranging from 1 cm/year to 15 cm/year [9,10] leaving the city more vulnerable to flooding and subsurface sea water intrusion. 

In regions of the world such as central Mexico that are experiencing rapid water table declines, this problem is met by constructing production wells that extend hundreds of meters below the present water table and with either open boreholes or perforated casings across their entire length [11,12]. These long-screened wells may simultaneously expose the population to anthropogenic contaminants such as nitrate (NO_3_) and geogenic contaminants such as arsenic (As) [11,13,14,15,16,17]. High concentrations of NO_3_ in excess of the World Health Organization (WHO) recommended limit of 10 mg/L NO_3_-N can kill infants under six months old when their formula is made with water with a high NO_3_ content [18]. In contrast, exposure to high As concentrations exceeding the WHO limit of 10 µg/L causes a wide range of serious and deadly diseases to people of all ages, which include many types of vascular diseases and cancers [19,20], as well as learning disabilities in infants and children that negatively impact cognitive function for life [21,22,23,24,25] and therefore lifetime earnings [12,26]. The mutual dilution of shallow and deep groundwater across such long-screened wells obscures the severity and risk of toxic concentrations of both anthropogenic and geogenic contaminants.

While the pump is off in these long-screened wells, the wellbores can act as highly permeable conduits through which non-negligible ambient flow passes from one layer with a higher hydraulic head to the other layer with a lower head. Ambient flow is defined as the vertical flow that occurs along a well that links at least two aquifers. This flow can be substantial even when the vertical head difference across the screened aquifers is very small (<1 cm) [27]. The prevalence of ambient flow was explored by Elci et al. (2001) [27], who reported significant ambient flow utilizing an downhole electromagnetic flowmeter in 104 of 142 tested wells at 16 sites across the United States. The observed ambient flow rates ranged from 0.01 L/min to 6.2 L/min. The screen length of these wells varied between 3.1 m and 56.6 m [27]. Detailed measurements of ambient flow were performed on a single well with a 76 m screen (66 m to 142 m below ground surface (mbgs)) utilizing an electromagnetic flowmeter in Western Australia by Poulsen et al. (2019) [28]. Although the vertical hydraulic gradient was only 8 × 10^−4^, a downward flow rate within the well of 1 L/min was observed between 66 and 77 mbgs, and this increased to 3.5 L/min between 123 and 130 mbgs. All ambient flow exited the well into the aquifer at the bottom (130–140 m) at a rate of 6 L/min [28]. Ambient flow even operates across wells with relatively short screen lengths. Ambient flow within a well with a 7.9 m screen length (9.5–17.4 mbgs) in Washington State, U.S., reached 4 L/min at 15.1 mbgs under a vertical gradient of only 3 × 10^−3^ [29]. 

Contaminants that are transported by the ambient flow can cross-contaminate the aquifers these wells are screened across [30,31,32,33,34,35]. Therefore, the ambient flow may bias the interpretation of the chemical and isotopic composition of water samples obtained from these wells when they are pumped. Even when steady state is reached in the values of physical-chemical parameters (pH, Specific Conductance (SC), temperature) before a water sample is taken, these groundwater samples are commonly interpreted to represent the chemical composition of a single chemically homogeneous aquifer, which could lead to bias in interpreting hydrogeochemical processes operating along local or regional flow paths within an aquifer. This is because ambient flow results in a mixture of the two or more distinct aquifer chemistries in varying proportions over hundreds of pumped wellbore volumes [34,36]. This may lead to bias that could impact results within or between studies with different sampling protocols. Goldrath et al. (2015) [37] analyzed dissolved As concentration in depth-dependent water samples collected from six depths spaced evenly across a 33 m screen. They found that As concentrations ranged from 6.4 to 17.6 µg/L and most of the As mass flux entered the well through a 16 m interval owing to both high As concentration and the high hydraulic conductivity (*K*) of this layer. Poulsen et al. (2019) [28] profiled chlorofluorocarbon (CFC) concentration in a well with a 76 m screen (66–142 mbgs) and found that the concentrations varied between 24.2 pg/kg and 55.6 pg/kg. The heterogeneity in the physical properties and chemical composition of the aquifer drives the transient chemical composition of produced water under the influence of ambient flow. Reilly and LeBlanc (1998) [38] demonstrated that the temporal variation in dissolved constituents in water samples collected from long-screened wells was impacted by two factors: (1) the physical and chemical heterogeneity of the aquifer; and (2) transport in the well. By using a numerical model McMillan et al. (2014) [32] found that the mixture of water from different depths in the produced water is influenced by the pump intake position, pumping rate and duration and wellbore flow rate. This demonstrates the risks from characterizing the chemical and isotopic composition of an aquifer by relying on one water sample collected from long-screened wells. 

Despite these complexities that have the potential to confound a simple interpretation of water samples taken from a long-screened production well, these wells remain the researcher’s primary source of insight into reaction and transport processes across a regional aquifer system. Numerous regional groundwater quality studies [39,40,41,42,43,44,45,46] have been published that relied on samples taken from long-screened wells, but these studies rarely mention the potential impacts of ambient flow. Authors rarely report whether temporal changes in chemical composition were observed during pumping or the duration the pumps had been turned off (or on) prior to sampling. Indeed, reference to the length of the well screens is rarely made. 

To control for the bias induced by ambient flow, depth-dependent sampling [35,47,48,49] or vertical monitoring well nests [28,34,50,51] have been employed in many studies. However, these methods are technically challenging and expensive. They may have limited potential to be scaled up to study hydrogeochemical processes within aquifers that are relied on for drinking water and food supply for much of the world’s population that lives in low- and middle-income countries. Aquifers that are crucial to the viability of regional economies and human existence commonly have water tables that lie hundreds of meters below ground surface within hard sediments or rock. This means that the cost of installing monitoring wells is prohibitive. Furthermore, the hydrogeologic information on these aquifers are commonly limited. Therefore, there is a need for affordable and technically feasible sampling and interpretation methods to detect and quantify ambient flow across the long-screened production wells. This can be used to correct groundwater quality bias induced by ambient flow and lead to additional insights into the sources of anthropogenic and geogenic contaminants within long-screened wells [11].

The transient chemical composition produced by a long-screened well provides an opportunity to gain insight into the ambient flow and chemical compositions of the layers of an aquifer that a well is screened across. The present study began with a survey of the steady-state chemical and isotopic composition of 19 production wells in the urban area of San Miguel de Allende (SMA), Mexico. Knappett et al. (2020) [11] confirmed that this aquifer has the highest concentrations of NO_3_ in the so-called Indepedence Basin. Some of these wells in the SMA area also have elevated concentrations of geogenic As and fluoride (F) which are thought to be related to the remnant geothermal heat beneath the city as it straddles the flanks of an extinct volcano and a major regional fault [11,15,17,52]. The spatial distribution of anthropogenically sourced NO_3_ and geogenically sourced As may be useful information for the city’s water supply utility to choose the location, depth and screened interval for new production wells in this area with rapidly falling water tables of 1–2 m/year [53]. 

In this study, we characterize the significance of ambient flow and the vertical stratification of groundwater chemistry. Seven wells that provide municipal water supply for SMA were investigated. Temperature and chemical variables were measured continuously using a calibrated multi-meter on the produced water during one day’s pumping. During the pumping, five discrete water samples were collected from each well to observe temporal changes in a wide range of ions, elemental composition and water isotopes. Radon (Rn) gas was also measured in two of the production wells in time series. Then, building on the classical end-member mixing analysis (EMMA) technique (i.e., [54]), groupings of variables that behave conservatively were used to identify the end-member chemical composition of a chemically stratified upper and lower aquifer. The determined end-member water chemistries were applied to quantify the volume and rate of ambient flow. To the authors’ knowledge, this approach has not been previously applied to calculate the ambient mixing of aquifers that have disparate chemistries through long-screen wells. Lastly, a revised conceptual model of the 3D distribution of water chemistry in the urban aquifer was produced through a joint analysis of the surface geology and land-use maps, the steady-state water chemistry from the 19 wells and the newly constrained end-member chemistries of the upper and lower aquifers. This low-cost approach may be useful in other regions to identify and isolate the independent contributions of surface-derived and geogenic contaminants in produced well water to inform management decisions in regions that are highly dependent on groundwater.

## 2. Materials and Methods

### 2.1. Study Area

The city of San Miguel de Allende (SMA) is located at the southern end of the semi-closed Upper Rio Laja Watershed which is also known as the Independence Basin [52]. This inter-montane volcano-sedimentary basin is located in northeast Guanajuato State on the northern edge of the Trans-Mexican Volcanic Belt (TMVB). The basin is surrounded by mountain ranges (Figure 1b). Normal faults and graben structures are widely present and are part of the extensive Taxco-San Miguel de Allende system of faults that run through central Mexico in a general NNE-SSW direction [55]. The only river within the basin is the Laja river, which flows intermittently into the Allende reservoir in the southwest corner of the basin (Figure 1b). Presently, surface water is scarce, and nearly all irrigation and municipal demand is met by groundwater pumping. A detailed trend analysis [53] performed on observed water levels in 61 production wells across the western half of the basin, known as the CARL aquifer (Cuenca Alto del Rio Laja), between 2008 and 2015 found that the water table declined at a rate of about 0.43 m/year. Li et al. (2020) [53] predicted that by the year 2035, 45% of the surface area of the CARL aquifer will have a water table deeper than 120 m. Below this depth, without major electricity subsidies for pumping, farming of most agricultural crops and standard irrigation techniques would no longer be profitable. The Mexican government, however, does heavily subsidize the cost of electricity for the agriculture sector by approximately 92% rendering the individual farm operations insensitive to this economic threshold [56]. 

San Miguel de Allende is the largest city in the basin with a population of 174,615 [16,57]. Underlying the city is the SMA fault (Figure 1c). This is a normal fault that forms a prominent 100 m cliff that cuts through the center of the city in the NNE-SSW direction. The aquifer to the west of the fault is designated the CARL aquifer, and the aquifer to the east is deemed the SMA aquifer by the state water agency (CEAG). The city’s potable water supply and sewerage utility (SAPASMA) treats the aquifers on both sides of the fault as two separate aquifers owing to different outcropping geology and vastly different water table elevations (Appendix A). Based on the basinwide mapping of water tables and groundwater chemistry, Knappett et al. (2018) [58] suggested that the SMA fault acts as a barrier to lateral mixing of groundwaters from the CARL and SMA aquifers, as the water table in the SMA aquifer is approximately 100 m higher than in the CARL aquifer (Appendix A and Figure 2a in Knappett et al. (2018) [58]). At the same time as acting as a barrier to the lateral movement of surficial groundwater, this fault may still act as a vertical conduit in places for rising, deep, geothermal waters [58]. In the vertical direction, the local aquifer system has been approximately mapped up to 400–500 m [11,16,58,59] depth, although reports of observed lithology during drilling of the wells are scarce. In general, the aquifer can be divided into the upper aquifer and the lower aquifer (Mahlknecht et al. (2006) [59], Figure 2 in Knappett et al. (2020) [11] and Appendix A in Knappett et al. (2022) [12]). The lower layer is a fractured ignimbrite aquifer consisting of mafic rocks, and the upper one is composed of poorly consolidated sedimentary deposits ranging from silt to boulder sized, with a high *K*. The upper aquifer serves as the main groundwater source and is generally where the water table is found [16,53]. In SMA and across the basin, aquifer properties (*K* and storativity (*S*)) have rarely been reported. Moreover, those reported values vary. CONAGUA (2020) [60] summarized pumping tests conducted in 2007 over 10 production wells located 6–10 km east of the SMA fault in the SMA aquifer which mostly comprises fractured andesitic volcanic rock and reported that *K* ranged from 0.2 to 10 m/day. Mahlknecht et al. (2006) [16] reported 5–430 m/day as the range of *K* for the aquifer in the SMA area based on results from pumping tests that were not described in any detail. To the authors’ knowledge, a detailed measurement of *S* has not been reported in any publications. Therefore, in the present study a pumping test was conducted 800 m west of the SMA fault within the sedimentary CARL aquifer to estimate aquifer properties (see more details in Section 2.4 and Appendix A). The SMA aquifer is mainly recharged by water from the surrounding mountains [58,59]. The rapid overexploitation creates the possibility of recharge from the downward flux of urban wastewater and irrigation return flow [58] and may enhance upwelling of geothermal waters through faults [11,53]. 

Within the SMA urban area, 7 wells (Figure 1c) were selected to investigate the impact of ambient flow within long-screened wells on the produced water quality and characterize the end-member chemistry of the upper and lower layers of the aquifer. These wells, referred to herein as time-series wells, are part of a network of 19 production wells that provide potable water to households of the city by SAPASMA (Figure 1c). Downhole video logs were provided by SAPASMA for all 7 time-series wells except Well 3. Well construction details for all wells are summarized in Table 1. These production wells were constructed from stainless-steel casings that are perforated over most of their length. The start of the perforated interval ranged from 1808 to 2007 masl for the 7 time-series wells. Their total perforated interval length ranged from 130 to 205 m. Surface geology and fault maps were obtained from [61,62,63,64] and edited based on the direct field observations by co-author Dr. Loza-Aguirre and published literature [16,64]. These observations were combined with borehole lithologies found in consulting reports [64,65,66] to constrain a 3D conceptual model of the urban area. A geological cross-section located just north of the study area is shown in Figure 2 in Knappett et al. (2020) [11]. 

Detailed geologic borehole lithologies were, unfortunately, not available for any of the 19 municipal wells that supply water to SMA, and therefore, the geologic layering could not be precisely determined in the study area. Visual inspection of well log videos obtained from SAPASMA, however, provided a clue for inferring the depth within the aquifer where important changes in Eh or pH occurred, through characterizing the presence and color of chemical precipitations of hydrous ferric oxides and mechanically accumulated particles [67,68,69,70]. For example, Figure 2 exhibits the transition from clean to clogged perforations in Wells 1 and 2 which were installed within the low-lying sedimentary CARL aquifer in the northern part of the city close to the SMA fault (Figure 1c). For both wells, which are located only 380 m apart and are screened across similar elevations within the aquifer (Table 1), the geochemical transition zone occurs at approximately 1735 m above sea level (masl). This transition was about 80 m below the static water table at the time of the recording of the well log video in 2010. This suggests the presence of two chemically distinct layers within the aquifer across the saturated thickness of the CARL aquifer. Unfortunately, such a clear transition from clean to clogged well perforations was not observable in other wells (Appendix A). 

### 2.2. Conceptual Model

Based on the geological information of the study area [11,53,59] and the evidence of transition points in the chemical weathering of the well casings, as well as the existence of pollutants with known surface and geothermal sources (i.e., the distribution of NO_3_ and As concentrations over 19 production wells in Appendix A panels d and k and Knappett et al. (2020) [11]), we propose a conceptual model with two chemically distinct aquifer layers (Aquifer 1 and Aquifer 2) across which the long-screened wells are perforated (Figure 3). In such a setting, Figure 3 illustrates the ambient flow in a long-screened well. The concentrations of a conservative groundwater constituent in Aquifer 1 and Aquifer 2 are designated as *C*_1_ and *C*_2_, respectively. Here, “conservative” means that there is negligible loss or production of that dissolved constituent as a result of physical decay or chemical reactions over the duration of time in which the pump is off (<12 h). Ambient flow in this setting is likely to be substantial as vertical head gradients in natural aquifer systems within inter-montane basins are commonly large [71,72,73,74]. Since the city is in a recharge zone, and the aquifer has been heavily pumped for many years, we propose that in SMA, Aquifer 1 is the upper chemically distinct layer and has a higher hydraulic head than Aquifer 2. Therefore, the ambient flow driven by the vertical hydraulic gradient will cause water to flow from Aquifer 1 to Aquifer 2 when the pump is turned off. This flow displaces and mixes with native groundwater in Aquifer 2 (Figure 3). The total volume of ambient flow flowing from Aquifer 1 to Aquifer 2 is termed *V*_AF_. 

### 2.3. Sampling of Production Wells

Amongst the 19 production wells that provide municipal water for SMA, seven of them were time-series wells (Figure 1c) that were studied to investigate the impact of ambient flow in the long-screened well on the temporally varying chemistry in the produced water. For these wells, temperature, SC, oxidative-reductive potential (ORP) and pH were monitored continuously using a calibrated multi-meter (YSI Professional Plus©, YSI Inc., Yellow Springs, OH, USA) during the pumping. Additionally, for each well, five samples were collected at different stages of the pumping, which were the first wellbore volume (within 5 min), 30 min, 90 min and 180 min after the start of pumping and within 5 min prior to the end of pumping. These samples were analyzed for alkalinity, anions, cations, water isotopes and, in the case of Wells 2 and 3, Rn. 

Alkalinity and Rn concentrations were measured on-site at the time of sampling. Alkalinity was measured by handheld titration with H_2_SO_4_ (Model AL-DT, HACH^©^). Rn concentrations were measured in Wells 2 and 3 using a RAD7 (Durridge Co., Inc., Billerica, MA, USA) portable radon-in-air monitor. Water samples for analyzing anions, cations and total dissolved elements were filtered through 0.45 μm nylon syringe filters (Ref: S25NY045, Simsii Inc., Irvine, CA, USA) into three pre-rinsed 20 mL high-density polyethylene (HDPE) scintillation vials (LS Vial, HDPE, Urea Cap Sep, PE Cone Lnr, Wheaton Industries Inc., Millville, NJ, USA). One vial was used to analyze for anions. Two vials were acidified to 0.1% nitric acid to analyze for cations and total dissolved elements. The nitric acid was distilled from 70% ACS grade nitric acid using an acid purification system (1000 DST, Savillex Corp., Eden Prairie, MN, USA). Lastly, one 20 mL glass vial was filled with filtered water to analyze for water isotopes (δ^18^O and δ^2^H). All samples were stored at 4 °C prior to analysis. 

Dissolved anions and cations were analyzed by ion chromatography (IC) (Dionex 500, Thermo Fisher Scientific, Waltham, MA, USA) at Texas A&M University (TAMU). Standards and laboratory blanks were measured every 15 samples to confirm accuracy. The analyzed cations include sodium (Na^+^), potassium (K^+^), calcium (Ca^2+^), magnesium (Mg^2+^) and lithium (Li^+^). The analyzed anions include chloride (Cl^−^), bromide (Br^−^), NO_3_^−^, sulfate (SO_4_^2−^) and fluoride (F^−^). Total dissolved elemental concentrations were analyzed by inductively coupled plasma mass spectroscopy (ICP-MS) (Element XR, Thermo Scientific, Waltham, MA, USA) at the Radiogenic Isotope Laboratory at TAMU. These elements include boron (B), iron (Fe), As, barium (Ba), phosphorous (P), silica (Si) and sulfur (S). Water isotopes (the ratios of δ^18^O and δ^2^H) were analyzed on a Picarro cavity ring-down system (Picarro Inc., Santa Clara, CA, USA) within the Stable Isotope Geosciences Facility (SIGF) at TAMU. Based on running the known standards from the International Agency for Atomic Energy (IAEA), Vienna Standard Mean Ocean Water 2 (VSMOW2) and Standard Light Antarctic Precipitation (SLAP), the error was estimated to be within ±0.1‰. 

In addition to the 7 time-series wells, the remaining 12 production wells (Figure 1c) were sampled to investigate the distribution of groundwater chemistry in the aquifer underlying SMA. For each well, one water sample was collected after the water chemistry (temperature, SC, ORP and pH) monitored by a calibrated multi-meter (YSI Professional Plus©, YSI Inc., Yellow Springs, OH, USA) became reasonably stable as assessed on-site. The collected water sample was analyzed for alkalinity, anions, cations and water isotopes utilizing the identical procedures detailed above. 

### 2.4. Pumping Test

Whereas single-well pumping tests have been performed on wells in the area [60], only transmissivity (*T*) can be estimated from a single-well pumping test. Furthermore, although single-well pumping tests provided *T* values for the SMA aquifer east of the SMA fault, no single-well pumping tests had been performed in the CARL aquifer near the city on the western side of the fault within the sedimentary material. Therefore, on 18 May 2018, a pumping test was performed utilizing an inactive production well (Well 1), located 380 m and 440 m away from two actively pumped production wells (Wells 2 and 3), as an observation well in the sedimentary part of the CARL aquifer (Figure 1c). This permitted the estimation of both *T* and *S*. Neighboring Wells 2 and 3 were simultaneously pumped for 4.5 h at a rate of 22.5 L/s and 3 h at the rate of 27.0 L/s, respectively. The changing water table in Well 1 was observed manually every 1 min utilizing a water level meter (Model 102, Water Level Meter, Solinst Canada Ltd., Georgetown, Ontario, Canada). To reduce interference in the water table by previous pumping, the pump in the observation Well 1 was turned off for 4 days prior to the pumping test. The steps followed to analyze the observed drawdown and derive values for aquifer *T* and *S* and corresponding discussions are included in Appendix A. 

### 2.5. End-Member Mixing Model 

Water pumped from a long-screened well is the mixture of water from aquifers that the well is screened over. During prolonged pumping, after steady-state conditions are reached, the concentration of a conservative constituent in the pumped water is *C*_steady_. Steady-state conditions are defined as the moment the absolute value of $\frac{dC}{dV}$ is less than 0.005, where *C* is the concentration (with respective units) and *V* is the pumping volume since the start of the pumping. The steady-state concentration is the average of the tracer’s concentration in each layer weighted for each layer’s transmissivity [34,51,75]. For a simple two-layer system, this can be expressed as:(1)$$T_{\mathrm{tot}}\;\cdot {\;C}_{\mathrm{steady}}{=T}_{1}\cdot {\;C}_{1}{+T}_{2}\;\cdot {\;C}_{2}$$
where *T*_tot_ is the total transmissivity of both aquifers (or layers) and *T*_1_ and *T*_2_ are the transmissivity of Aquifer 1 and Aquifer 2, respectively. Based on the proposed conceptual model (Figure 3), the unpumped wellbore acts as a conduit for ambient flow from Aquifer 1 to Aquifer 2. Therefore, water sampled from the first wellbore volume would comprise water from Aquifer 1, and the concentration of a conservative constituent in this water would be *C*_1_. The value *C*_steady_ can be directly observed. Based on Equation (1) and known values of *C*_1_ and *C*_steady_, *C*_2_ can be expressed as:(2)$$C_{2}=\frac{T_{\mathrm{tot}}}{T_{2}}C_{\mathrm{steady}}-\frac{T_{1}}{T_{2}}C_{1}$$

In SMA, the upper and lower aquifers are considered hydraulically connected, since the hydraulic heads of wells screened across one or both aquifers are similar [53]. Therefore, the upper and lower aquifers, which have a similar *K* value, are treated as one single hydraulically connected aquifer. Based on this characteristic and the equation that *T* = *K*·*B*, where *B* is the saturated thickness of the aquifer, Equation (2) can be simplified to:(3)$$C_{2}=\left(1+\frac{B_{1}}{B_{2}}\right)C_{\mathrm{steady}}-\frac{B_{1}}{B_{2}}C_{1}$$
where $B_{1}$ and $B_{2}$ are the saturated thickness of Aquifer 1 and Aquifer 2, respectively. Therefore, once the thickness ratio of the two aquifers is obtained, the value of *C*_2_ can be solved from Equation (3). This method was inspired by the end-member mixing analysis (EMMA), which was originally developed to determine the percentage of streamflow derived from different water sources with distinct chemical compositions [76,77]. To determine which dissolved ions or elements behave conservatively, bivariate plots are made of the concentrations of every dissolved constituent against that of every other constituent for all water samples in the study area [77]. For the aquifer that was conceptualized as having two chemically distinct layers (Aquifer 1 and Aquifer 2), the concentrations of conservative constituents in the pumped water should form a straight mixing line when plotted against other conservative constituents in bivariate plots. This was used as a test to identify which parameters behaved conservatively and had different values in Aquifers 1 and 2.

### 2.6. Calculating Ambient Flow

When two layers are chemically distinct (*C*_1_ ≠ *C*_2_) with respect to a specific conservative constituent, the ambient flow causes the concentration of that constituent in pumped water to vary until the steady state is reached (Figure 4) [32,34,78]. This characteristic can be used to estimate *V*_AF_. If ambient flow is absent, the produced water chemistry will not change during pumping, and each constituent will be persistently *C*_steady_. This is represented by the blue line in Figure 4. Under this assumed scenario, the total mass flux of a conservative constituent, indicated by the gray area in Figure 4, is *V*_tot_ ·*C*_steady_, where *V*_tot_ is the total volume of pumped water during the pumping. In the presence of ambient flow, however, as shown in Figure 4, the actual total mass flux (*M*_tot_) during the pumping is the area bounded between the horizontal axis and the observed, black curve, which can be obtained from actual measurements. The absolute difference between the two types of total mass flux, indicated by the red dashed area in Figure 4, is caused by the ambient flow, which replaces water in Aquifer 2 that has concentration *C*_2_ with the water from Aquifer 1 that has concentration *C*_1_. Therefore, the absolute difference in mass flux caused by the aforementioned replacement is $V_{\mathrm{AF}}\cdot \left|C_{1}-C_{2}\right|$. Based on the above statement, the mass flux balance can be expressed as:(4)$$V_{\mathrm{AF}}\cdot \left|C_{1}-C_{2}\right|=\left|M_{\mathrm{tot}}-V_{\mathrm{tot}}\;\cdot {\;C}_{\mathrm{steady}}\right|$$

Then, *V*_AF_ can be calculated as:(5)$$V_{\mathrm{AF}}=\left|\frac{M_{\mathrm{tot}}-V_{\mathrm{tot}}\;\cdot {\;C}_{\mathrm{steady}}}{C_{1}-C_{2}}\right|$$

If the duration that the well was turned off prior to the start of pumping (*t*) is known, the average ambient flow rate (*Q*_AF_) over the entire well screen is
(6)$$Q_{\mathrm{AF}}=\frac{V_{\mathrm{AF}}}{t}$$

## 3. Results

### 3.1. Temporal Variation in Produced Water Chemistry

At the city and basin scale, high As concentrations in well water are associated with a high temperature and high Na concentrations (Appendix A and Knappett et al. (2020) [11]). High As concentrations are also associated with low Ca and NO_3_ concentrations (Appendix A). Moreover, high NO_3_ concentrations are associated with high Cl, Mg and Ca (Appendix A). At the single-well scale, under the impact of ambient flow, the concentration of conservative constituents pumped through a well screened over two chemically distinct layers will temporally change until all ambient flow is removed. Based on the bivariate plots (Appendix A), the conservative constituents which differed in their concentrations across Aquifer 1 and 2 included temperature, SC, Cl, NO_3_, Na, Ca, Mg, S, Ba, B, As and Rn. The variations in Na concentration, temperature and SC in the 7 time-series wells are presented to illustrate the evidence for ambient mixing (Figure 5). Sodium concentrations increased during pumping in all wells except Well 7, but the magnitude of those changes varied between wells. For example, Na concentration increased from 73.9 to 85.0 mg/L, from 68.5 to 74.5 mg/L, from 91.8 to 97.1 mg/L and from 144.6 to 152.9 mg/L in Wells 2, 3, 4 and 5, respectively. In Wells 1 and 6, however, Na concentrations increased only slightly from 99.7 to 103.6 mg/L and 83.1 to 84.7 mg/L, respectively. In Well 7, Na concentration decreased slightly from 67.7 to 65.5 mg/L. Temperature consistently increased during pumping in all seven wells. Temperature increased substantially by 3.5 °C, 3.2 °C and 2.2 °C in Wells 2, 3 and 4, respectively. A smaller increase was observed in Wells 1, 5, 6 and 7 of 0.3 °C, 0.4 °C, 0.6 °C and 1 °C, respectively. 

In contrast to Na and temperature, SC decreased during pumping. Specific conductance decreased the most in Wells 2 and 4 from 827 to 590 µS/cm and from 818 to 670 µS/cm, respectively. This is a decrease of 237 µS/cm and 148 µS/cm, respectively. For Wells 1, 3, 5, 6 and 7, the SC decreased by 27 µS/cm, 41 µS/cm, 55 µS/cm, 39 µS/cm and 24 µS/cm, respectively. The concentration of all measured constituents (conservative and non-conservative) during the pumping is detailed in Data Availability Statement.

Based on the observed variation in conservative constituents, the pumped volume or the number of wellbore volumes needed to reach a steady state can be estimated. For each well, concentrations of most conservative constituents were only observed a maximum of five times. Temperature and SC were measured every 5 min. Such sparse measurements on their own were inadequate to calculate the time or pumped volume that steady-state chemistry was reached. This pumped volume when steady-state is reached is needed to calculate the value of *M*_tot_ for use in Equations (4) and (5) to estimate *V*_AF_. Therefore, a curve was first fit to each tracer to smooth the temporal concentration changes. The fitted curves for the observed Na concentrations are presented to illustrate how good this fit typically was (black solid line in Figure 5). The root mean square error (RMSE) between observed and best-fit Na for seven wells was always <0.1 mg/L (Figure 5). Therefore, the best-fit curves accurately represented the temporal variation of observed concentrations. The goodness-of-fit statistics (RMSE and R^2^) for all conservative constituents in all seven wells are detailed in Appendix A and Appendix A. 

As shown by the vertical dashed lines in Figure 5, the pumped volume required to reach steady-state varied between wells. Generally, the required pumped volume ranged between 200 m^3^ and 300 m^3^. For Well 7, however, only 100 m^3^ of pumped groundwater volume was required to reach the steady state. Furthermore, for each well, the estimated pumped volume to reach steady-state depended varied with the conservative constituent. The variation was commonly modest. For example in Well 1, the required pumped volume estimated from temperature, SC and Na was 200 m^3^, 250 m^3^ and 230 m^3^, respectively. In Well 5, however, the same tracers revealed a required pumped volume of 210 m^3^, 230 m^3^ and 360 m^3^, respectively.

### 3.2. Calculating End-Member Chemistries of Aquifers 1 and 2

To estimate the temperature and chemical composition of conservative constituents in Aquifer 2 utilizing Equation (3), the values of *B*_1_ and *B*_2_ (or the ratio of *B*_1_/*B*_2_) need to be obtained or estimated first. This refers to the dimensions of the chemically distinct layers which are difficult to constrain without independent observations from vertical nests of short-screened monitoring wells or vertical profiling using packer tests or a similar technique within the long-screened wells [28]. In this case, we utilized the observed transition from clean to the presence of chemical precipitates on the inside of the perforated casing in Wells 1 and 2 (Figure 2) to estimate the thickness of each layer. The reasoning is that the precipitates form at a depth in the aquifer when the Eh or pH changes which is likely to be the dividing depth between the chemically distinct layers. Using this method of inference, the ratio of *B*_1_/*B*_2_ was 1/1.2 and 1/0.7 in Wells 1 and 2, respectively. These are close to even. Owing to the lack of observed differences in well clogging over depth in other wells (Appendix A), *B*_1_/*B*_2_ was set to 1 for all wells for the following analysis. In each well, the water chemistry reached a steady state before the last water sample was collected (Figure 5). Therefore, the concentration of the conservative constituents in the last water sample is representative of *C*_steady_. 

The calculated concentrations of conservative constituents in Aquifer 1 and Aquifer 2 for all seven wells are presented in Figure 6. Since Rn concentration was only measured in Wells 2 and 3, it is presented in Appendix A. Generally, Aquifer 1 has higher SC, Cl, NO_3_, Ca, Ba and Mg than Aquifer 2. In contrast, Aquifer 2 has a higher temperature, Na, B and Rn. The aquifer with the highest concentrations of S and As varies between wells. In the case of S, the two aquifers have similar concentrations except for in Wells 2, 3 and 4. In Wells 2 and 3, the concentration of S is higher in Aquifer 2. In contrast, in Well 4, S concentrations are lower in Aquifer 2 compared to Aquifer 1. Arsenic concentrations in Aquifer 2 in Wells 1–4 are higher than in Aquifer 1. However, in Wells 5–7, As concentration in Aquifer 2 is slightly lower than in Aquifer 1. The differences in the values of conservative constituents in each Aquifer were typically greater between wells than between aquifers. For example, both aquifers in Wells 5 and 7 have the highest and lowest SC, respectively (Figure 6e,g). 

To test the impact of the ratio of *B*_1_/*B*_2_ on the calculated water chemistry in Aquifer 2, the concentration of conservative constituents in Aquifer 2 was also estimated when *B*_1_/*B*_2_ equaled 0.7 and 1.2, respectively. As shown in Appendix A, the variation of the estimated chemical composition in Aquifer 2 was negligible when the ratio of *B*_1_/*B*_2_ varied. 

### 3.3. Calculating Ambient Flow Rate

Based on the best-fit curve (Section 3.1) and Equation (5), *V*_AF_ can be estimated from each conservative constituent (Appendix A). The median values of *V*_AF_ across all conservative constituents for Wells 1 through 7 were 170.6 m^3^, 65.1 m^3^, 65.6 m^3^, 31.8 m^3^, 60.3 m^3^, 46.7 m^3^ and 27.8 m^3^, respectively (Figure 7). Calculated *V*_AF_ varied the greatest across the different constituents in Wells 1 and 2. These varied from 14.4 m^3^ to 354.7 m^3^ and 3.8 m^3^ to 346.8 m^3^, respectively. For the studied wells, SAPASMA documented the average daily pumping hours for each month in 2018 (Appendix A). All sampling for this study was conducted in May 2018 during which the daily unpumped hours for Wells 1–7 were 12.6, 8.3, 11.8, 14.7, 11.7, 8.7 and 14 h, respectively. Therefore, based on Equation (6), the median ambient flow rates for Wells 1–7 were 225.7 L/min, 130.8 L/min, 92.7 L/min, 35.9 L/min, 85.8 L/min, 89.4 L/min and 33.1 L/min, respectively. 

## 4. Discussion

### 4.1. Estimating Ambient Flow Rate 

Based on the temporal variability in the values of conservative constituents during the pumping and the proposed method in Section 2.6, the ambient flow rates (*Q*_AF_) of Wells 1–7 ranged from 33.1 to 225.7 L/min. These rates are higher than some measured values (e.g., 0.01–6.3 L/min from Elci et al. (2001) [27], 4 L/min from Vermeul et al. (2011) [29] and 1–7 L/min from Poulsen et al. (2019) [28]). This discrepancy raises the question of whether the proposed method can yield reasonably accurate estimates of *Q*_AF_. To test the proposed method, we applied it to analyze the data presented in Figure 8 in Mayo (2010) [34]. This study reports the temporal variation of total dissolved solids (TDS) in a well (SW-89) with a 13 m screen and that was pumped for 23 days. The well investigated in Mayo (2010) [34] was unpumped for 71 days before the start of pumping and screened across two chemically distinct layers of an aquifer, which had native TDS concentrations of 128 mg/L and 5193 mg/L. During pumping, the pumping rates declined from 12.7 L/s to 9.1 L/s [34]. However, the author did not report the detailed pumping schedule. Therefore, we used 12.7 L/s and 9.1 L/s to calculate the maximum and minimum pumping volume, respectively, which can derive maximum and minimum *Q*_AF_, respectively. Based on the aforementioned information, the *Q*_AF_ derived from Equation (6) ranged from 13.9 L/min to 19.5 L/min. Unfortunately, Mayo (2010) [34] did not measure or estimate *Q*_AF_ of this well. Therefore, we were not able to compare *Q*_AF_ calculated by an independent approach to that calculated using our approach. However, the author did estimate *Q*_AF_ for a well (SW-67) that had a similar length of the well screen and was located 10.5 km away from SW-89. The estimated rate of SW-67 ranged from 0.6 to 2.4 L/min [34]. The lithology at SW-67 was dominated by finer-grained sediments, whereas SW-89 has coarser-grained sands [34]. Therefore, the *K* value near SW-89 was expected to be higher than near SW-67. Additionally, the vertical head difference between stratifications that SW-89 screened across was about 1.2 m, which was 10 times higher than the head difference (0.13 m) presented in SW-67. Therefore, the *Q*_AF_ in SW-89 is expected to be larger than in SW-67. Thus, our method produced reasonable results when applied to another study with a much longer time period in which the pump was turned off.

Poulsen et al. (2019) [36] demonstrated that *Q*_AF_ is related to aquifer properties (*K* and vertical anisotropy) and vertical head gradients. Therefore, we suspected that the high estimated values of *Q*_AF_ in our study might be caused by the relatively large *K* value and vertical head gradient owing to the mountainous setting and the high rate of pumping beneath the city. According to the analysis of the pumping test that we performed in the vicinity of Wells 1–3, the representative *K* value for the aquifer in the vicinity of these wells in the CARL aquifer was 57 m/day or 93 m/day (Appendix A). These two values were derived from best-fit numerical and analytical models, respectively (Appendix A). The range of *K* values for the reported *Q*_AF_ in Elci et al. (2001) [27] (0.01–6.3 L/min) and Poulsen et al. (2019) [28] (1–7 L/min) was 0.4–50 m/day and 19–170 m/day, respectively. Therefore, the estimated *K* in the vicinity of Wells 1-3 in the present study was higher than most of the wells analyzed for ambient flow in the aforementioned studies, except for one well location (170 m/day) in Poulsen et al. (2019) [28]. Therefore, it is reasonable that *Q*_AF_ would be higher in our study area. 

In this study, the estimated value of *Q*_AF_ is mathematically related to the estimated value of *V*_AF_ (Equation (6)). The estimated *V*_AF_ is not expected to vary with the conservative constituent utilized. However, the ranges of calculated *V*_AF_ were in some cases relatively large. Most notably in Wells 1 and 2, they ranged from 14.4 to 354.7 m^3^ and 3.8 to 346.8 m^3^, respectively. Additionally, as shown in Appendix A, the values of *V*_AF_ estimated from the temporal variation of Ba were much higher than the value estimated from other conservative constituents over four wells (Wells 1 and 4–6), suggesting systematic bias. In Wells 2, 3 and 7, a large *V*_AF_ was estimated from NO_3_, Cl and S, respectively (Appendix A). These large variations might be induced by uncertainties in estimating the natural concentration of each tracer in each aquifer layer [34]. Therefore, an accurate estimation of the water chemistry of each stratification would improve the estimation of *Q*_AF_.

### 4.2. Implications for Best Practices for Sampling Long-Screened Wells 

From a practical point of view, long-screened wells are utilized to study the pore-water chemistry of the aquifer as well as the quality of the water used for drinking or irrigation [39,40,41,42,43,44,45,46]. Although the length of the well screen is commonly not reported in studies that utilize such wells, wells that serve municipal, industrial and agricultural needs commonly are built with long screens to maximize production. Therefore, we suspect that ambient flow as well as the weighted averaging of the chemical composition of two or more aquifers that may or may not have reached steady-state influence the results of many of these studies. Few such studies, however, mention a sampling tool or protocol aiming at reducing the impact of ambient flow on the analysis of aquifer chemistry other than monitoring temperature, SC, ORP and pH to ensure stability. In this study we demonstrate that ambient flow through the well causes the chemical composition of produced water to vary with time and volume pumped (Figure 5). The changes are gradual enough that they would not be noticeable if they were being monitored on-site. The water chemistry continued to change beyond 40 wellbore volumes. Therefore, the timing of sampling will impact the analyzed groundwater quality (Figure 4 and Figure 5). As shown in Figure 5, the values and concentrations of parameters vary until the *V*_AF_ is completely purged. The pumps in the seven time-series wells studied here were turned off for 8–15 h prior to pumping. The wells then required 2–4 h (15–40 wellbore volumes) of pumping to remove *V*_AF_ (Figure 5). In Wells 2 and 3, the temperature difference between the beginning and the end of the test is approximately 3 °C. In Well 4, the Na concentration increased by approximately 5 mg/L during the test. Therefore, if the water sample is collected before the water quality is stabilized, the analyzed water chemistry will be unduly weighted toward the chemistry of a high-head layer that the long well screen intercepts. 

The time or purged volume needed for the water quality to stabilize varies site by site. Mayo (2010) [34] pumped two long-screened wells, which had not been pumped for 123 and 71 days, respectively. In those cases, water chemistry stabilization was not achieved until 8.5 and 15 days, respectively [34]. Poulsen et al. (2019) [36] developed a numerical model to estimate the volume needed to be purged to completely remove *V*_AF_ (i.e., reaching a steady state) and found that the purged volume ranged from 2.2 to 20.6 times the *V*_AF_ under differing physical conditions of an aquifer such as varying *K* values and heterogeneity. In this study, approximately 300 and 100 m^3^ were purged to remove *V*_AF_ in Wells 2 and 7, respectively (Figure 5). 

After the *V*_AF_ is fully purged, the pumped water is the mixture of water from screened aquifers weighted for the *T* of each chemically distinct layer. As shown in Figure 5, the stabilized temperature in Well 4 was 27.7 °C. However, the estimated temperature in Aquifer 1 and Aquifer 2 that Well 4 was screened across was 25.6 °C and 29.4 °C, respectively (Figure 6). Similarly, in Well 2, the steady-state Na concentration was 85 mg/L, and the estimated Na concentrations in Aquifer 1 and Aquifer 2 were 75 mg/L and 89 mg/L, respectively. Mayo (2010) [34] analyzed the concentration of TDS in a long-screened well that screened over two chemically distinct aquifers. The steady-state concentration of TDS was 3530 mg/L. This amount differed greatly from TDS in the two layers which were 10,400 mg/L and 702 mg/L, respectively [34]. This demonstrates that the water chemistry of pumped water from a long-screened well can lead to a misunderstanding of the actual water chemistry in the aquifer, the processes contributing to it and the health risks of consuming it. 

### 4.3. Health Concerns with Falling Water Tables in Aquifers with Anthropogenic and Geogenic Contaminants 

The falling water tables in SMA (Table 1) cause a decrease in the saturated thickness of Aquifer 1. In future decades, since wells must be deepened to continue to produce water at a high enough rate to satisfy demand, a greater proportion of groundwater from Aquifer 2 will be pumped. In this region with shallow geothermal heat, this presents specific health risks [12,79]. Currently, the steady-state As concentration in water from Well 2 within the CARL aquifer is approximately 18 µg/L (Appendix A), which was below the Mexican guideline of 25 µg/L [80] but above the WHO limit (10 µg/L). As shown in Figure 6, for Well 2, the estimated As concentrations in Aquifer 1 and Aquifer 2 are 13.9 µg/L and 22.7 µg/L, respectively. When more water from Aquifer 2 is pumped, the steady-state As concentration in the pumped water will increase. The increase in As concentration is likely driven by the dissolution of silicate minerals, which is accelerated by hot, geothermal water [11,81]. As shown in Figure 6, Aquifer 2 has a higher temperature than Aquifer 1. Therefore, a falling water table will drive more mixing with water from Aquifer 2, which will increase the As concentration. Chronic consumption of drinking water with high As concentrations drives higher rates of cardiovascular diseases, cancers and diabetes in adults and infant mortality and depresses cognitive development in children [19,24,82,83,84,85]. The fact that Wells 5–7 show overall lower As concentrations and a lack of stratification (depth trend) suggests the southern part of the city on the flanks of the Palo Huerfano Volcano will produce higher-quality water for more years to come than the lower-elevation CARL aquifer which has a higher temperature and geogenic As.

Radon gas concentrations in Aquifer 2 were also higher than in Aquifer 1 (Appendix A). For instance, in Well 2, Rn gas concentrations in Aquifers 1 and 2 are 952 and 4,153 pCi/L, respectively. With the falling water table in Aquifer 1, more water from Aquifer 2 containing a higher Rn concentration will be extracted. Radon present in household water represents a health risk to humans through inhalation rather than ingestion [86]. The WHO does not provide a Rn standard for drinking water because it is considered more appropriate to measure Rn concentration in indoor air than in drinking water because the main entry pathway into humans is through the lungs, not the stomach [86]. The WHO standard for Rn concentration in indoor air is 100 Bq/m^3^ (2.7 × 10^3^ pCi/m^3^) [86]. It has been estimated that drinking water with 1000 Bq/L (2.7 × 10^4^ pCi/L) Rn can increase the Rn concentration by 100 Bq/m^3^ in indoor air [86]. Therefore, pumping more water from Aquifer 2 may result in local people being exposed to a high Rn environment and cause health concerns, since Rn is a carcinogen and is the second leading cause of lung cancer following smoking [87,88]. Interestingly, in several cases, the water sampled from municipal and rooftop storage tanks had much lower Rn concentrations than sample directly from the well head from which the water was sourced (data not shown). This suggests that the common act of storing water in a tank on the rooftop of homes will allow Rn to degas and experience radioactive decay (half-life of Rn is 3.8 days), making it much safer for household use.

In terms of NO_3_ concentration in public drinking water supply, the WHO guideline is 50 mg/L (bulk NO_3_), which is set to protect against infant methemoglobinemia [18]. For Well 3, the estimated NO_3_ concentration in Aquifer 1 was 61 mg/L (Figure 6), which exceeds the WHO guideline. However, the NO_3_ concentration in Aquifer 2 was only 15 mg/L (Figure 6). The water from Aquifer 2 dilutes the NO_3_ concentration in pumped water and results in the steady-state NO_3_ concentration of 30 mg/L (Appendix A) which satisfies the WHO guideline. This dilution effect was also found at the basin scale with respect to the mutual dilution of waters high in As and NO_3_ [11], but here we quantified the transport and mixing processes that account for the negative correlation in their concentrations seen at the basin scale. With the falling water level in Aquifer 1, the steady-state NO_3_ concentration will likely decline but at the expense of increasing As concentrations with an attendant rise in the incidence of disease.

### 4.4. Limitations of This Study

The certainty of the results presented in this study is limited by two factors: limited lithologic data and the absence of depth-dependent water chemistry data. Other field sites, however, may not have these limitations and may be able to use the methodology developed and tested here to achieve more accurate results. High-quality geologic borehole data are commonly unavailable in many parts of the world that are most in need of groundwater monitoring where rampant dewatering of aquifers in agricultural areas occurs. We were unable to accurately determine the thickness of Aquifer 1 and Aquifer 2. We relied on the first appearance of chemical clogging observed by a downhole camera to estimate the thickness in two wells. Consequently, when the estimated thickness was applied to Equation (3), this increases the uncertainty in the estimated water chemistry of Aquifer 2. This lack of data may not hinder the application of the proposed method in places where more information is available on the dimensions of the layers with distinct chemistry. For example, aquitards commonly separate upper aquifers, contaminated by urban or agricultural sources, from lower aquifers [34,89,90]. In such cases, it may be obvious what the dimensions and even *T* are of each aquifer layer that a well is screened across. When the thickness of Aquifer 1 and Aquifer 2 can be obtained, the water chemistry in Aquifer 2 (i.e., the low-head aquifer) can be better constrained. 

The most reliable approaches to determine the water chemistry of different stratifications are depth-dependent sampling along the continuously screened well (i.e., [28]) and vertical nests of monitoring wells that are screened across short intervals at each depth (i.e., [34]). However, these approaches are not always possible since they require a great deal of technical capability, heavy equipment and expense, especially when the water table lies hundreds of meters below the surface as is the case in this basin in central Mexico. In this study, we presented a rapid, inexpensive method to identify and then estimate the strength of chemical stratification and the volumetric flux that is permitted by inter-connections across chemically distinct layers created by the presence of the long-screen well. Evidence like this can be used to build a case for a more thorough investigation into the spatial distribution of anthropogenic and geogenic contaminants within potable water aquifers. 

Due to the absence of actual measurements of water chemistry in Aquifer 1, we used the first water sample to represent the water chemistry of Aquifer 1, which was employed in Equations (2) and (3) for the estimation of water chemistry in Aquifer 2. This assumption is valid when there are only two chemically distinct aquifers. When there are more than two, the initial water produced from the well is the mixture of water from multiple aquifers, except the one with the lowest hydraulic head [28,34,91]. Under this condition, the water chemistry of the first water sample cannot represent any single aquifer. This means that Equations (2) and (3) would be unsolvable without more information. Although the proposed method only works for a simple aquifer system, the low cost and straightforward equations that require minimal information to solve for the end-member chemistries and ambient flow make this approach useful in a wide number of aquifers that are impacted by both surface and geogenic contaminant sources in the deeper aquifer. 

## 5. Conclusions

An analysis of the time series of the conservative chemical composition and temperature in water pumped from seven long-screen municipal production wells in the city of San Miguel de Allende revealed they source water from chemically distinct aquifers. This analysis also revealed the end-member concentrations of those constituents that temporally varied during the pumping in the upper and lower aquifer. The pumping time or wellbore volume required for the water quality being stabilized varied over wells, which were 2–4 h (15–40 wellbore volumes). Therefore, the timing of a water sample taken may greatly impact the analyzed groundwater temperature and chemistry. After the produced water chemistry stabilizes, the natural concentration of each tracer in each aquifer can be estimated if the transmissivity of the aquifers is known. A broader application of the approach presented here to study areas with both surface-derived and deeper subsurface-sourced contaminants will improve the understanding of the vertical changes in groundwater quality with depth and thereby help cities and communities plan where to place the screens of new production wells so that they are more likely to produce safe drinking water. Long-screen production wells facilitate the mixing of shallow water containing anthropogenic contaminants with deep water containing geogenic contaminants, which in the short term has the effect of reducing the concentrations of each.

## Figures and Tables

**Figure 1 ijerph-19-09907-f001:**
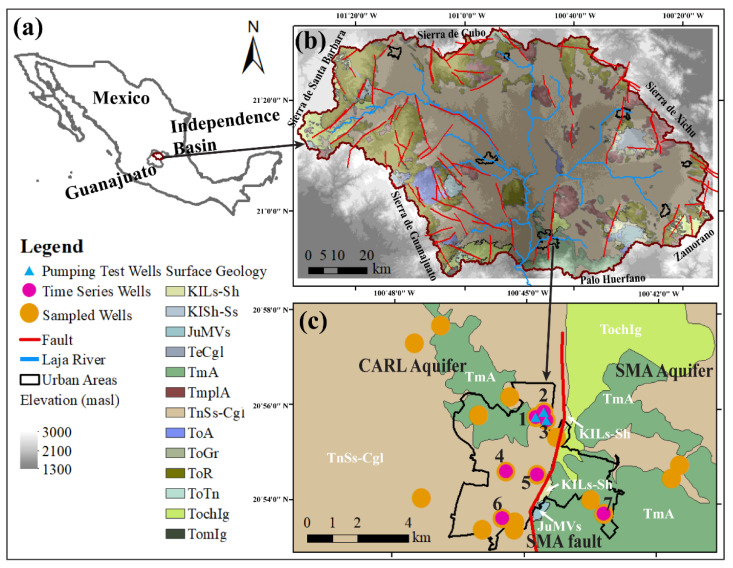
(**a**) Location of Guanajuato State and the Independence Basin. (**b**) The distribution of surficial rock or sediment type and deposition age, topography and the locations of major faults and urban areas within the basin. masl is meter above sea level. (**c**) Surface geology and location of production wells (large orange circles) within the SMA urban area. Seven pink circles indicate wells monitored continuously during a one-day pumping. Three blue triangles indicate wells used for the pumping test. The SMA fault cuts through the city trending in the NNE-SSW direction (solid red line). The surface geology symbols mean: KILs-Sh: Lower Cretaceous Metamorphic Limestone; KISh-Ss: Lower Cretaceous Metamorphic shale interbedded with metamorphic sandstone; JuMVs: Lower Cretaceous Meta-volcanosedimentary; TeCgl: Tertiary Eocene polymictic conglomerate; TmA: Tertiary Miocene Andesite; TmplA: Tertiary Miocene–Pliocene Andesite; TnSs-Cgl: Tertiary Neogene interbedded Sandstone and polymictic Conglomerate; ToA: Tertiary Oligocene Andesite; ToGr: Tertiary Oligocene Granodiorite; ToR: Tertiary Oligocene Rhyolite; ToTn: Tertiary Oligocene Tonalite; TochIg: Tertiary Oligocene Chattian Ignimbrite; TomIg: Tertiary Oligocene–Miocene Ignimbrite.

**Figure 2 ijerph-19-09907-f002:**
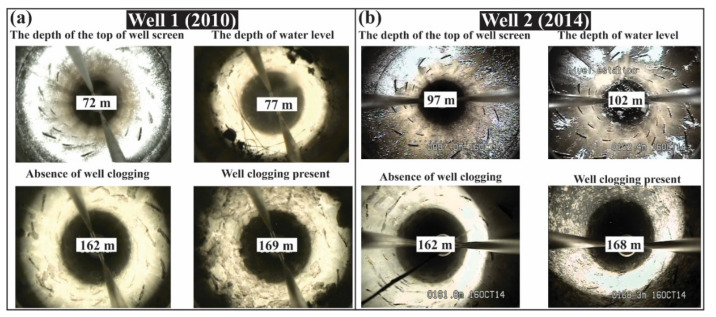
Snapshots of well log videos recorded with downhole camera in Wells 1 (**a**) and 2 (**b**), respectively. The parentheses next to the well ID indicate the year that the well log video was recorded.

**Figure 3 ijerph-19-09907-f003:**
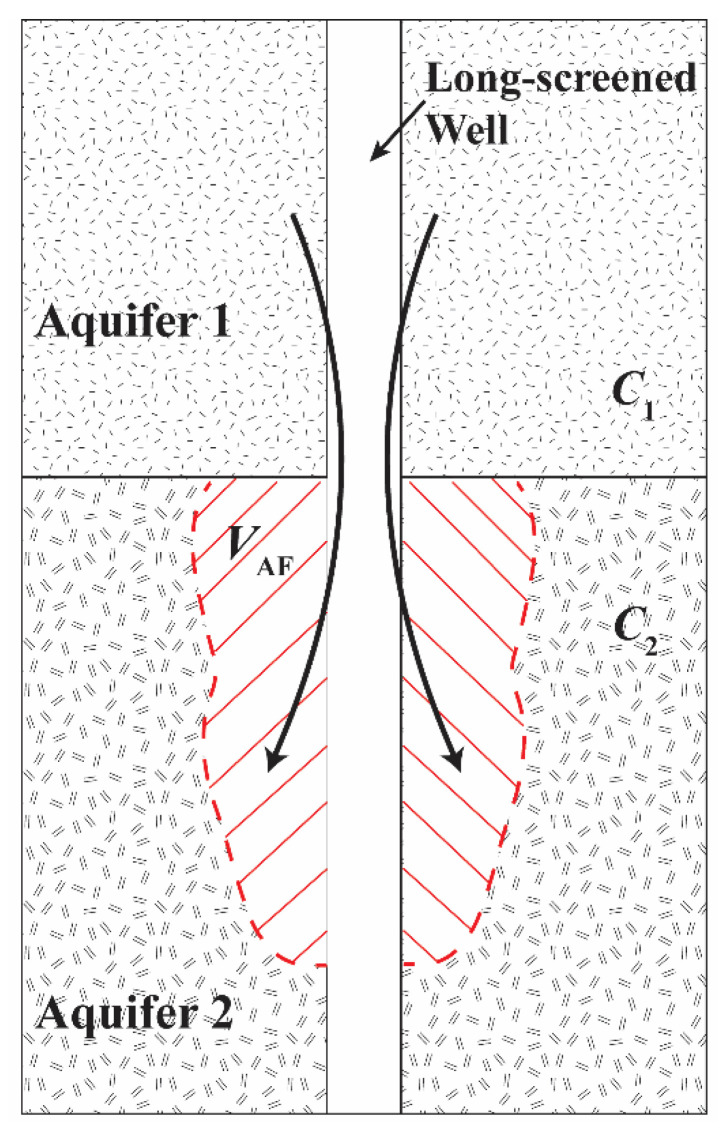
Conceptual model of ambient flow from high hydraulic head Aquifer 1 to low hydraulic head Aquifer 2.

**Figure 4 ijerph-19-09907-f004:**
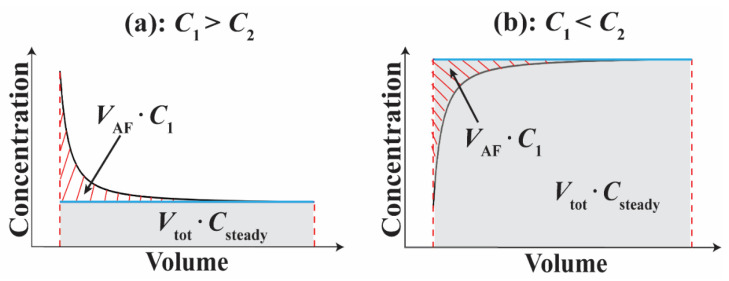
Specific scenarios generating changes in the concentration of a conservative tracer during pumping: (**a**) *C*_1_ > *C*_2_ and (**b**) *C*_1_ < *C*_2_. In each panel, the black curve indicates the observed concentration during the pumping. The blue line indicates the concentration under the steady state.

**Figure 5 ijerph-19-09907-f005:**
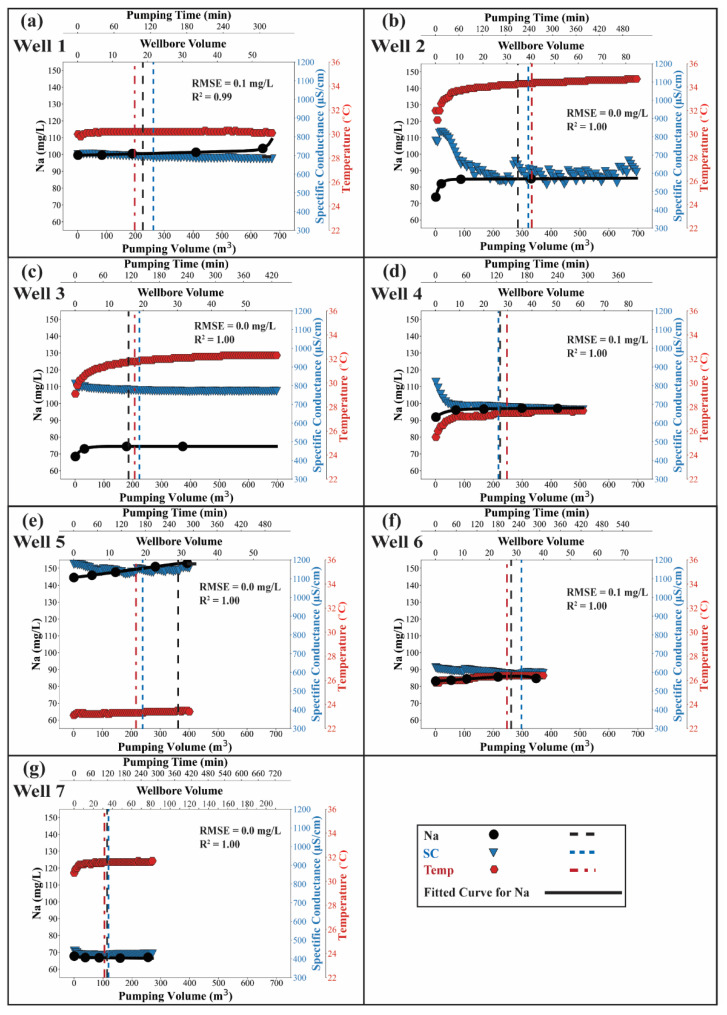
Variation in sodium (Na) concentration, specific conductance (SC) and temperature (Temp) in produced water from 7 time-series wells. Panels (**a**–**g**) represent Wells 1–7, respectively. For each well, the black solid line indicates the fitted curve for the observed Na concentration. The root mean square error (RMSE) and the coefficient of determination (R^2^) between observed and fitted Na concentrations are presented in each panel. The black, blue and red dash lines indicate the timing that Na, SC and Temp reached steady state, respectively.

**Figure 6 ijerph-19-09907-f006:**
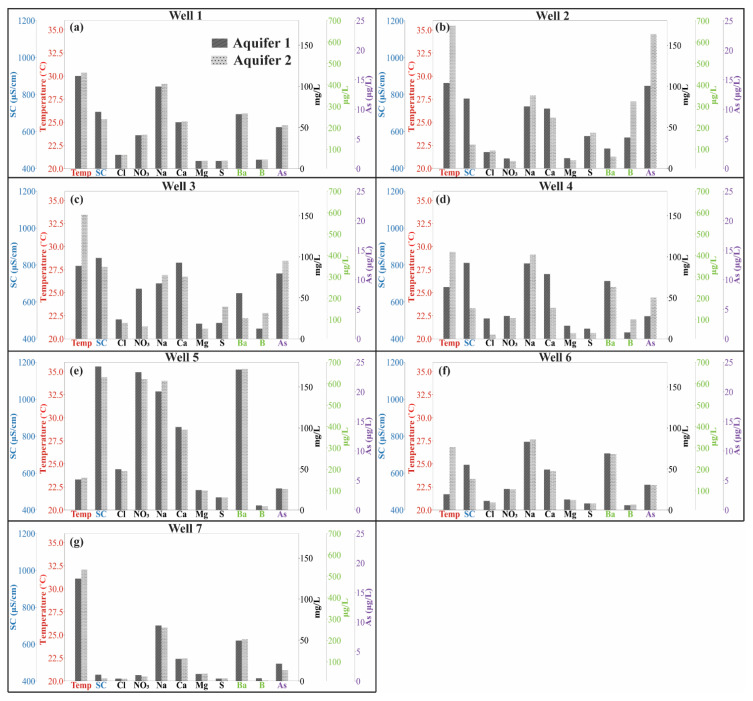
The water chemistry of conservative constituents (temperature, SC, Cl, NO_3_, Na, Ca, Mg, S, Ba, B and As) in Aquifer 1 and Aquifer 2 over 7 wells. Panels (**a**–**g**) indicate Wells 1–7, respectively. The color of tracer’s name on the horizontal axis matches the color of its corresponding vertical axis.

**Figure 7 ijerph-19-09907-f007:**
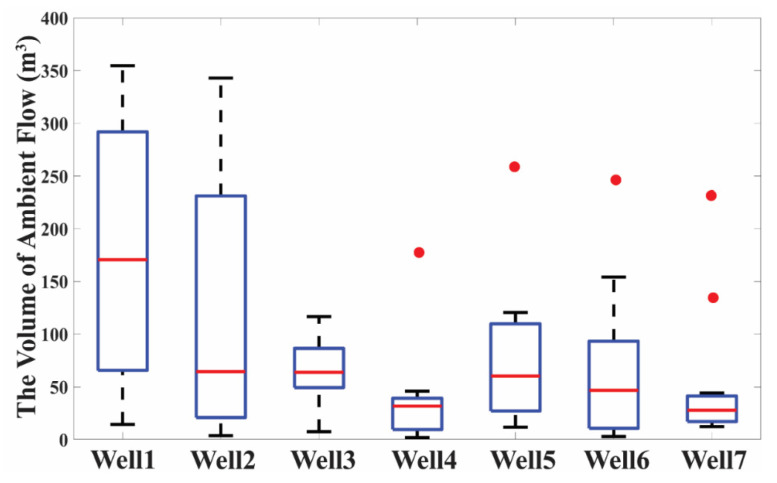
The estimated volume of ambient flow (*V*_AF_) in 7 wells. Red point represents the outlier values of calculated ambient flow volume, which is larger than 1.5 times the interquartile range above the third quartile.

**Table 1 ijerph-19-09907-t001:** Well construction, static water level and pumping rates for the 7 municipal production wells of SMA.

Well ID			1	2	3	4	5	6	7
Well Top Elevation (masl ^1^)			1892	1904	1919	1879	1903	1939	2081
Diameter of Well (cm)			30	24	29	30	30	30	29
Pumping Rate (L/s)			35.0	22.5	27.0	29.3	23.0	20.0	16.1
Perforated Interval (masl)		Top	1820	1812	na ^2^	1822	1808	1843	2007
		Bottom	1622	1607	na	1656	1653	1713	1836
Observed Groundwater Level (masl)	Year		2010	2014		2006	2011	2016	2014
	From downhole camera		1815	1802	na	1803	1834	1846	1892
	Year		2018						
	From SAPASMA		1793	1789	1801	1774	1829	1845	1883

^1^ masl: meter above sea level. The elevation was measured by a handheld GPS (etrex 30, Garmin). ^2^ na: not available.

## Data Availability

The data including the observed drawdown from the pumping test and continuous measurements of temperature, SC, pH and ORP are available at http://www.hydroshare.org/resource/f823835ae4294cedb8dcd860d95cc6d4 (accessed on 21 June 2022). The analyzed water chemistry of discrete water samples collected from seven time-series wells is included in Appendix A in the Appendix A.

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
