# Peer review of "Water Quality Assessment Bias Associated with Long-Screened Wells Screened across Aquifers with High Nitrate and Arsenic Concentrations"

_ijerph, 2022, doi:10.3390/ijerph19169907_

Round 1
Reviewer 1 Report
1. Introduction
In the introduction, a good link is made between previous similar research and what is new in this approach. This is also supported by an adequate number of references.
What is missing in the introductory part is a link to the hydrogeological basis of the researched area, which is certainly important when it comes to abstract water from the underground (aquifers).
2. Materials and Methods
Line 220-223 and 227-228 Missing a hydrogeological profile from which it could be clearly visible.
Line 249-250 This is quite a significant shortcoming, because if there is no hydrogeological profile and no geological data, then it seems to me that the conclusions about what are the natural factors (especially arsenic) that affect water quality are only drawn based on chemical analyses. That is not good, because then we know what present situation is , but we don't know exactly what causing it. It should be clarified why detailed geologic borehole lithologies are missing and how this affects the quality of the estimates.
Line 251-254 I think it would be good to include some of the recordings from the Supplementary in the main text itself.
Figure 1 It needs to be improved. I don't know if this is a geological map?? If so, why are the faults not red? Urban areas can be marked with a black line. A map can really be problematic when it is black&white. On the map c), surface geology labels could be added directly to polygons with labels.
The coordinates along the edge b and c of the sketch are missing, which would allow for precise placement in space. The graphic scale should have at least two or three divisions.
The altitude scale in this form is not needed - you just need to put a color scale with the minimum and maximum altitude or consider simply adding isohypses instead of shading the altitudes.
Line 274 Typo ertiary > Tertiary
Table 1 Unify the graphics of the table. It doesn't look good like this (lines of different thicknesses). And there is too much additional interpretation along with the table (counted exponents) - it would be good to combine part of it in the names of columns/rows, and transfer part of it to the main text before or after the table. Also my question why mbgs is used? Why didn't you just convert everything to masl, because it is hard to keep track in this way?
Line 291-293 A reference needs to be added.
Figure 3 Description is too long for image. The text up to the first fullstop can remain as the description of the picture, and everything after that can go into the main text with a reference to picture 3.
3. Results
Figure 4 Same comment as in Figure 3
4. Discussion
The arguments are coherent, balanced and compelling. The authors also pointed out possible shortcomings of the study, which is certainly a good thing. But it seems to me that this can also be a serious problem when making relevant conclusions. According to the authors themselves, they lack:
- precisely determined thickness of Aquifer 1 and Aquifer 2
- geologic borehole data
- the absence of actual measurements of water chemistry in Aquifer 1
- the ambient flow is from multiple aquifers and the proposed method only works for a simple aquifer system
5. Conclusion
The conclusions are partially supported by the results presented in the article. There are no additional references n secondary literature.
References
The article is adequately referenced.
Supplementary
Figure S1 Faults should be red, and urbanization could be marked with some other colour. Coordinates along the edge of the map that would allow precise placement in space are missing. I like this map better than the one in Figure 1, so you should think about displaying it in the main text, not in the supplementary
Line 120-163 Why is some of this text not incorporated into the main text in the part where the conditions in the aquifer are described?
English language and style
I don’t feel qualified to judge about the English language and style because I am not a native English speaker, but everything seems understandable to me and I feel that no major interventions in the text are needed.
Reviewer 2 Report
The manuscript is devoted to the analysis of data from wells drilled and screened in different aquifers. It is well known that the use of wells screened at different depths might enhance the transport of contaminants. This can be related to leakage through aquitards (I myself studied this problem in my MSc thesis, 35 years ago; a scientific publication was derived from that work: https://doi.org/10.2307/3430650) or to vertical flow in wells.
The work is generally well done and the manuscript is quite comprehensive. Also the language and the quality is sufficiently clear.
In my opinion, the main scientific weakness is the fact that in section 2.5 End-member mixing model, the physical approximations on which the model is based are not clearly discussed. For instance, equation (1) assumes that the hydraulic gradients in the two aquifers assume the same value.
SPECIFIC COMMENTS
1) The term "ambient flow" is used in different situations in hydrology. Therefore, I suggest to give a more precise and explicit definition of this term and to clearly link it to "vertical flow in well".
2) At line 141, "ambient" and "active" mixing is mentioned. The difference between the two concepts (ambient vs. active) should be clearly defined.
3) Line 377. Which are the measurement units of dC/dV? They should be added to 0.005. How is this threshold chosen?
TECHNICAL COMMENTS
1) Line 37. Expression "an urban aquifer that overlies remnant geothermal heat" should be rephrased.
2) Line 135. Why no reference is given here, if "hundreds of regional water quality studies have been published"?
3) Table 1. The line "Year 2018" is confusing, because 2018 appears under the column of well 4 only.
4) Line 378. The "pumping volume" should be explicitly defined.
5) When ranges of values are given, point 7 of the check list at page v of the NIST Guide for the Use of the International System of Units (SI) (https://physics.nist.gov/cuu/pdf/sp811.pdf) should be rigorously followed.
Round 2
Reviewer 1 Report
Dear authors,
thanks for the answers and clarifications sent.
I understand the reasons why you did not present the geological and hydrogeological data in more detail, and I thank you for clarifying this further in the text. Although it is still a missing part for me, I understand that the paper can present the obtained results with quality even without that part, and I am inclined to attribute this to author's freedom and give the green light to publish the work.
Author Response :
Thanks for coming up with concerns about the missing of geological and hydrogeological data. In San Miguel de Allende (SMA), the rapid decline of water level and the deterioration of groundwater quality has addressed more and more attentions. Therefore, groundwater study is significant for the local development. However, the lack of hydrogeological data may induce certain uncertainties into these studies. With more studies dedicated to investigating the SMA or the entire Independence Basin, we anticipate that more hydrogeological data will be obtained in future, which will enhance the understanding of local aquifer system.
Thanks again for your time and valuable comments for improving this manuscript.
Thanks,